

# Expediting evidence synthesis for healthcare decision-making: exploring attitudes and perceptions towards rapid reviews using Q methodology

Shannon E. Kelly[1,2], David Moher[1,3] and Tammy J. Clifford[1,4]

[1] School of Epidemiology, Public Health and Preventive Medicine, University of Ottawa, Ottawa, Ontario, Canada
[2] Cardiovascular Research Methods Centre, University of Ottawa Heart Institute, Ottawa, Ontario, Canada
[3] Centre for Practice Changing Research, Ottawa Hospital Research Institute, Ottawa, Ontario, Canada
[4] CADTH, Ottawa, Ontario, Canada

Corresponding author
Shannon E. Kelly,
skell102@uottawa.ca

## ABSTRACT

**Background:** Rapid reviews expedite the knowledge synthesis process with the goal of providing timely information to healthcare decision-makers who want to use evidence-informed policy and practice approaches. A range of opinions and viewpoints on rapid reviews is thought to exist; however, no research to date has formally captured these views. This paper aims to explore evidence producer and knowledge user attitudes and perceptions towards rapid reviews.

**Methods:** A Q methodology study was conducted to identify central viewpoints about rapid reviews based on a broad topic discourse. Participants rank-ordered 50 text statements and explained their Q-sort in free-text comments. Individual Q-sorts were analysed using Q-Assessor (statistical method: factor analysis with varimax rotation). Factors, or salient viewpoints on rapid reviews, were identified, interpreted and described.

**Results:** Analysis of the 11 individual Q sorts identified three prominent viewpoints: Factor A cautions against the use of study design labels to make judgements. Factor B maintains that rapid reviews should be the exception and not the rule. Factor C focuses on the practical needs of the end-user over the review process.

**Conclusion:** Results show that there are opposing viewpoints on rapid reviews, yet some unity exists. The three factors described offer insight into how and why various stakeholders act as they do and what issues may need to be resolved before increase uptake of the evidence from rapid reviews can be realized in healthcare decision-making environments.

## INTRODUCTION

The requirement for timely input to policy and healthcare decision-making encouraged evidence producers to accelerate their processes, resulting in the approach often referred

to as a rapid review. Rapid reviews expedite the evidence synthesis process by streamlining or tailoring the rigourous and explicit methods of a systematic review, and a variety of approaches may be employed (*Ganann, Ciliska & Thomas, 2010*; *Hailey et al., 2000*; *Harker & Kleijnen, 2012*; *Hartling et al., 2015*; *Khangura et al., 2012*; *Khangura et al., 2014*; *Polisena et al., 2015*; *Tricco et al., 2016*; *Tricco et al., 2015*; *Featherstone et al., 2015*). Use of rapid reviews is seemingly increasing as organizations struggle to meet the needs of knowledge users requesting timely evidence-informed decision support. Research on rapid reviews is consequently expanding in parallel as investigators endeavour to fill documented knowledge gaps by attempting to define and validate approaches and quantify the implications and impact of use (*Ganann, Ciliska & Thomas, 2010*; *Peterson et al., 2016*). Despite these efforts, little is known about the opinions or views of both evidence producers and knowledge users towards rapid reviews.

Although informal opinions about rapid review approaches seem to abound, no formal study of these views exists in the literature. Early on, as this type of approach was new to the evidence synthesis landscape, there appeared to be a certain stigma attached to rapid reviews. The general perception at that time was that a rapid review was a "quick and dirty" systematic review, and there was pushback from conservatives in the research methodology community despite uptake from knowledge users. Previous work has highlighted common themes associated with rapid reviews: the aim to inform health care decision-makers (also called knowledge users), a deficit in reporting and transparency of conduct, and an unclear, heterogeneous of approaches that all fall under the broad umbrella term 'rapid review' (*Coates, 2015*; *Moher, 2015*; *Schünemann et al., 2007*; *Tricco, Antony & Straus, 2015*). Yet, despite some central shortcomings, rapid response programs continue to thrive and expand internationally (*Hartling et al., 2015*; *Polisena et al., 2015*). Although negative views may persist, it seems clear that there is a distinct group of supporters who continue to cautiously champion and encourage the use of rapid reviews to support evidence-informed decision making. No research to-date has attempted to capture these views, and we know very little empirically how producers subjectively feel about rapid reviews and the risk of flawed conclusions or 'less than best' data that may result from methodological concessions made while accelerating the evidence synthesis process. Similarly, no organized study to-date has measured the end-users opinions on rapid reviews, how they value this form of evidence, or their experience with rapid reviews in practice.

Though difficult to delineate, improved understanding of the prevailing or varying values, beliefs and attitudes pertaining to rapid reviews is essential, as mindset can influence action, conduct and uptake. Investigating attitude and perception is important for a few reasons. *Cross (2005)* states that attitude and opinion help to form cognitive relationships, which in turn may influence actions or conduct. In fact, attitude can be defined by a predisposition to act in a particular way (*Ajzen & Fishbein, 1972*). As such, positive thought may lead a producer or user to approach or value rapid reviews in a positive manner or predispose them to behave in a supportive fashion. Likewise, more cautious, or even pessimistic notions around rapid reviews may influence conduct, curb use, or sully their value which may limit impact. Comparing and contrasting these

factors among and across the various stakeholder groups may serve to identify correlations in thought, similarities in experience or gaps in needs or methods. In addition, a comprehensive understanding of beliefs towards rapid reviews will further inform both evidence producers and researchers on how to continue to support the needs of decision-makers.

## A broader view of rapid reviews

This study is part of a broader research program on rapid reviews which involved three independent, yet related studies, including: 1) a scoping review of rapid review samples to map characteristics and methods of rapid reviews and to check their adherence to conduct and reporting guidelines (*Kelly, Moher & Clifford, 2016*); 2) a modified-Delphi study aiming to identify key defining characteristics of rapid reviews (S. Kelly, 2016, unpublished data); 3) the Q methodology study on attitudes and perceptions reported here. The course of action from the associated study program contributed to the sharing of knowledge and provided a gateway for an expanded discourse on rapid reviews; however, opening the dialogue and merely summarizing the collective thought is insufficient. *Hartling et al. (2015)* note the importance of studying end-user perspectives in their recent work. In order to further our global understanding of rapid reviews, it is also important to study the attitudes and traits of those who produce them. Given the absence of evidence from either perspective, a more formal review is merited.

## Q methodology

Fundamentally, Q methodology is a research methodology that allows for the systematic study of subjectivity (*Brown, 1993*). The method employs both qualitative and quantitative methods to reveal and detail viewpoints, values, attitudes, and opinions among a group of participants on a particular topic (*Watts & Stenner, 2012*). A few key objectives underlie the use of a Q methodology; First, the goal of finding the range of communicated ideas on a topic, followed by exploration of the prevailing variations in it, and finally, to logically connect these variations in an orderly way to each other. Following individual rank-ordering of statements, connecting of viewpoints is completed statistically through an inversion of conventional factor analysis (*Watts & Stenner, 2012*). The difference lies in the assessment of correlation of individuals rather than tests or mathematical variables. Correlation is done across viewpoints and ultimately mapped to results labelled 'factors.' Following a careful and methodological interpretation, the resulting factors represent the participants' subjectivity on the topic, and tell a specific 'story' about their beliefs, values and perceptions (Fig. 1) (*Brown, 1993*).

There are a small number of fundamental steps essential to the Q methodology (*Watts & Stenner, 2012*). First, collection of a broad, balanced sample of statements, referred to as the 'concourse,' which represents all relevant dialogue about a topic of interest. The concourse is then further refined into a set of statements called the 'Q set.' The Q set broadly represents the opinion field for the topic described in the concourse and is balanced to ensure that individual items capture each idea without gaps or

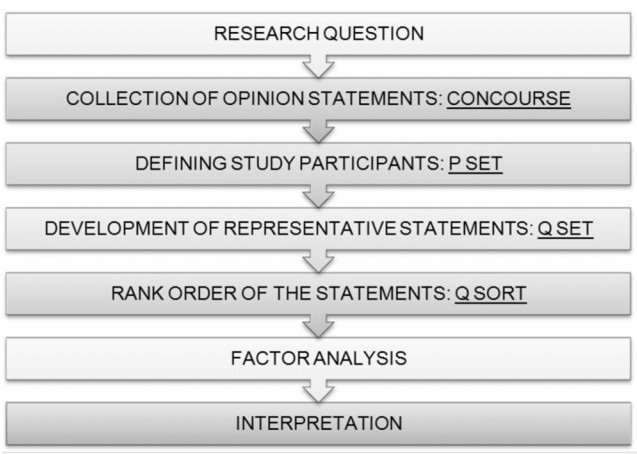

**Figure 1 Steps in Q methodology.** (*Cross, 2005*).

unnecessary overlap. It is important to note that balance in this sense does not mean that half of the statements are positive and the other half negative.

There is no solitary or exact way to produce a Q set. According to theory, it must be "tailored to the requirements of the study and the demands of the research question it is seeking to answer" (*Akhtar-Danesh et al., 2011*). Development of the Q set can generally be done in a *structured* or *unstructured* manner. In a structured Q set, relevant subject matter is organized into themes or ideas based on research or observation or possibly, a preconceived theory. Items in the Q set are then generated to ensure that all relevant themes identified are covered. An unstructured Q set is also constructed based on the entirety of relevant themes and key ideas collected. The more flexible sampling process allows for arrangement of the Q set into a series of statements representative of the whole population in the concourse. Arguably there is more freedom with this method of Q set definition; however, efforts must be made to maintain a rigourous and comprehensive process. All Q sets are ultimately judged on comprehensiveness, representativeness and balance in relation to the research question while remaining unbiased to any particular viewpoint (*Harker & Kleijnen, 2012*).

Next, a set of participants is selected who are referred to as the P set. Q methodology allows for use of small sample size as the aim is not to estimate population statistics, but rather to maximize diverse viewpoints and show the existence of these views (*Brown, 1993*). Q-sets of between 40 and 80 items have become standard as anything less may not represent the views or be too limiting (*Curt, 1994*; *Rogers, Stenner & Gleeson, 1995*).

Participants are asked to review the Q set of randomly ordered statements and to rank each from those they agree with the most to those they disagree with the most. This sorting is done using a pre-defined grid based on a quasi-normal distribution. This is a standard means of simplifying the following statistical procedure without biasing the factors that are interpreted (*Brown, 1980*). Both online and mail-in approaches have been validated against traditional in-person interviews and sorting. Studies have shown reliability and validity, with no difference in outcomes associated with changing the method of administration (*van Exel & de Graaf, 2005*). Online methods also allow

for access to a theoretically relevant sample of individuals that have a much broader national or international distribution. Although Q-methodology gives the impression that it is a difficult methodology for a lay-person to self-administer, studies have shown that parents, clinicians and the general public respond well when instructions are clear (*Akhtar-Danesh et al., 2011*; *van Exel & de Graaf, 2005*). This method is referred to as 'Q sorting.' Following the Q sort, the data are analysed using factor analysis. Finally, factor interpretation is carried out based on the results of the analysis phase.

The main goal of this study is to gauge how producers and knowledge-users feel about rapid reviews, and explore the range of opinion on this useful evidence synthesis approach. In addition, this study hopes to gain an understanding of whether these types of evidence summaries are valued, appropriate, or if certain misgivings persist given the lack of validation against gold standard methods. This paper presents the results of an explorative Q methodology study of evidence producer and knowledge user attitudes and perceptions towards rapid reviews.

## MATERIALS AND METHODS

Q methodology was used to identify participants' collective attitudes and perceptions towards rapid reviews.

### Setting

Web-based Q sorting using Q-Assessor software with e-mail communication to participants (*The Epimetrics Group, 2016*).

### Defining the Q set

A broad, comprehensive concourse representing a range of views towards rapid review was derived from a wide-range of sources prior to the refinement of the Q set (*Ganann, Ciliska & Thomas, 2010*; *Harker & Kleijnen, 2012*; *Khangura et al., 2012*; *Gough, Thomas & Oliver, 2012*; *Hailey, 2009*; *Watt et al., 2008a*; *Watt et al., 2008b*). The goal at this stage was to document as many viewpoints as possible pertaining to rapid reviews, before refining these views into the list of opinion statements that form the Q set. Following this approach, statements for the concourse were gathered or extracted using: 1) a small set of academic papers focused on rapid reviews reviewed in detail for author viewpoints; 2) conference proceedings where opinions and beliefs towards rapid reviews were transcribed in-person from plenary, oral or panel presentations, and posters or abstracts were examined for relevant opinions (questions or comments related to expediting evidence production during these proceedings were also transcribed); 3) detailed review of online resources, social media (Twitter) and electronic mailing lists and, 4) in-person consultation in the form of unstructured dialogue with experts in evidence synthesis, including both users and producers of rapid reviews. The first author attended the 2011 to 2014 Canadian Agency for Drugs and Technologies and Health (CADTH) Symposiums, the 2013 International Cochrane Colloquium in Quebec City, Quebec, and the 2014 Cochrane Canada Symposium in Ottawa, Ontario in order to document and record concourse statements.

A semi-structured approach was used to reduce the total number of viewpoints in the concourse to a balanced and representative Q set, eliminating repeated thoughts and redundancies. Items, or individual opinion statements, were organized based on similar themes and viewpoints pertaining to rapid reviews, including: timing, scope, perceived or actual bias resulting from tailoring methods, transparency or reporting, inconsistency in approaches used to expedite evidence synthesis, ways of tailoring methods, breadth or depth of evidence collected, utility of rapid reviews in decision-making, appropriateness, validity of results, confidence in results, and suggestions for future research in this area. Best practices for evidence synthesis (the gold standard for synthesizing evidence) were also factored in to the Q-set statements from the concourse (rigour, precision of estimate, reproducibility, transparency, explicit methods). Views in the concourse from both producers and knowledge users also reflected general support for the use of rapid reviews as well as hesitance or reserved judgement until further research is published. Statements from the concourse were refined based on these themes and viewpoints into a Q set of 50 items. This process was carried out by the first author in consultation with the second and third authors.

For the purposes of this study, the following terminology was used: 1) *Evidence producers* are anyone who carries out research activities and may be involved in evidence synthesis activities in any capacity; 2) *Knowledge users* are those who are likely to use the information generated through research to make informed decisions about health policies, programs, and/or practices; 3) *Systematic reviews* employ rigorous methods, such as those prescribed by the Cochrane Collaboration, to summarize research evidence; 4) *Rapid reviews* generally follow similar systematic methods, but these methods are tailored or modified in order to synthesize research evidence in a more timely manner to meet the particular needs of knowledge users; and 5) *Health technology assessment* is a broader, policy-based assessment of medical, social, ethical and economic implications of development, diffusion and use of health technologies.

## Assembling the P set

In order to maximize the possibility that a variety of perspectives could be articulated, and to ensure that both knowledge users and producers were captured in the sample, participants in this study were identified using a purposeful sampling approach. Study authors used a publicly available list of attendees present at four consecutive years of the annual CADTH symposium to identify both producers and knowledge users. The goal was to capture participants actively involved in the areas of policy making, program decision-making, healthcare delivery and research. Ineligible attendees were removed, including those who were industry representatives, administrative staff, students, patient group association representatives and those who attended the annual symposium only once in the four years reviewed. The initial recruitment goal was set at 50 participants.

Representatives were then split into two distinct lists representing knowledge users and producers and randomly ordered. Starting with the first person in each ordered list, those without a publicly accessible email address were removed in order to be consistent with Canada's anti-spam legislation that came into effect July 1, 2014 (http://fightspam.gc.ca).

A large proportion of CADTH research staff were included in the sample. As such, the proportion of researchers was restricted to no more than 20% CADTH staff in order to maintain an unbiased grouping. After obtaining approval from the Ottawa Hospital Research Ethics Board (protocol #20120143-01H), the first 25 individuals in each list were invited to participate using standardized email invitations containing a letter outlining the study objectives, methods, expected time commitment, consent information, and a link to the online Q sort. Replacement invitations were sent to new potential participants following a refusal or when no response was received from an invitee after two reminders sent at seven day intervals.

The final P set consisted of 11 participants (53 invitations, response rate 20.8%). Five invitations were declined for unspecified reasons, four email invitations bounced and two potential participants contacted the first author to decline for personal reasons, including lack of time (n = 1), lack of knowledge on rapid reviews (n = 1). A profile of the expert panel is provided in Table 1. Participants were predominantly female (72.7%) with doctoral degrees (63.6%) and aged 35 or older (81.8%). All were from Canada. Two knowledge users, eight producers and one respondent who did not consider themselves part of either category responded. Names of participants have not been identified in order to maintain the anonymity agreed upon as a condition of participation. Participants were not compensated.

Although the sample size may be considered small for a conventional factor analysis, the fundamental ideology and statistical technique underlying the Q method maintain that as long as a potential range of views are covered, small sample sizes may be adequate for the level of understanding sought out (*Watts & Stenner, 2012*).

### Q sort table

A predetermined grid, or Q sort table, based on a quasi-normal distribution was used (Fig. 2). The table consisted of 50 cells spread across seven columns of varying length matching the number of statements in the Q set. Scores of +3 (*most agree*) and −3 (*most disagree*) were assigned to anchor the extreme limits of the Q sort table, with sequential label numbering heading each column, including a value of '0' for neutral.

Five individuals piloted the instructions, Q set and Q sort online using Q-Assessor, a web-based software dedicated to Q methodology that was customized for this study (*The Epimetrics Group, 2016*). Instructions were examined for clarity, feasibility, and to ensure that the practical elements of design were appropriate. Q set was examined by the same individuals and the study team for content validity. Feedback did not result in a reduction or increase in the number of statements, only minor adjustments to wording for clarity or ambiguity.

### Q sort process

Participants completed the online Q sorting process using *The Epimetrics Group (2016)*. On the initial page, they were provided with an overview of study terminology, reminded of the study objectives, and asked to proceed to a preliminary Q sort of the 50 statements on rapid reviews. Next, participants sorted the statements into broad categories of "agree,"

**Table 1 Profile of the producers and knowledge users (n = 11).**

| Geographic location | n (%) |
|---|---|
| Canada | 11 (100) |
| **Age:** | **n (%)** |
| 18 to 35 | 2 (18.2) |
| 36 to 50 | 4 (36.4) |
| 50 or above | 5 (45.5) |
| **Sex:** | **n (%)** |
| Female | 8 (72.7) |
| Male | 3 (27.3) |
| **Education:** | **n (%)** |
| Doctorate | 7 (63.6) |
| Masters | 3 (27.3) |
| Undergraduate | 1 (9.1) |
| **Consider themselves:** | **n (%)** |
| Researcher/Producer | 8 (72.7) |
| Knowledge user | 2 (18.2) |
| Neither | 1 (9.1) |
| **Have ever been the author of a rapid review** | **n (%)** |
| Yes | 7 (63.6) |
| No | 4 (36.4) |
| **Have used a rapid review to aid in a policy or decision-making?** | **n (%)** |
| Yes | 7 (63.6) |
| No | 3 (27.3) |
| Unsure | 1 (0.9) |

| Least Agree | | | | | | Most Agree |
|---|---|---|---|---|---|---|
| | | **Statement Scores (n)** | | | | |
| **-3** | **-2** | **-1** | **0** | **1** | **2** | **3** |
| (2) | (6) | (10) | (14) | (10) | (6) | (2) |

**Figure 2 Fixed distribution for the Q-set.**

"disagree" and "neutral" based on their own understanding, opinions and perceptions of rapid reviews. In the subsequent step, the Q sort table was presented (Fig. 2). Participants were first asked to consider the range of opinion represented in the Q set while selecting and placing the statements they most strongly agree (n = 2) or disagree (n = 2) with onto the extremes of the table. After that, they free-sorted the remaining 46 statements, moving and reviewing the position of each until satisfied that they best reflected their own views. Finally, participants were asked to qualify selection of the four strongest views ranked in their Q sort by providing a succinct explanation for their choices in a free-text

comment box. A brief profile questionnaire was also administered. Based on pilot testing, it was estimated that the time to completion would be between 20 and 25 minutes. Actual time to completion was 45 min, on average.

### Analysis and factor interpretation

Q sorts entered by participants were analyzed using by-person factor analysis (i.e., statistical analysis is based on the individual, instead of a statement, characteristic or trait) in Q-Assessor. Initially, a correlation matrix was created to identify patterns of agreement and disagreement across the individual Q sorts. Correlations larger than 1.96 times the standard error ($1/\sqrt{n}$, where n is the number of statements) were used to identify significant relationships in the data (*Brown, 1993*). Factor extraction was initiated by calculating centroid factor loadings from the data. Positive and negative associations between each Q sort and the seven preliminary factors were explored. Eigenvalues, variance (total and by factor) and communality ($h^2$) were also calculated. Eigenvalues and individual variance represent the strength of the factor extracted and its potential explanatory power, with a higher value representing superior factor choices (*Watts & Stenner, 2012*). Communality is a measure of how much an individual Q sort holds in common with the other sorts in the study and is used for comparing and contrasting individual response across the initial (unrotated) factors.

Following a careful assessment of the preliminary factor loadings across a number of potential factor arrangements and their associated statistics, primary factors were extracted. In order to be considered, factors had to have an eigenvalue $\geq 1.00$ and at least one significant loading as assessed by the *Fuerntratt (1969)* criterion following a varimax rotation of latent factors. Additionally, the group of residual factors had to account for at least 40% of the total variance in the Q sorts (*Kline, 2014*).

Prior to factor interpretation, factor arrays (Table 2) and normalized weighted average statement scores (z-score), or factor scores, were calculated. Statements with a significant factor score ($p < 0.05$) were considered distinguishing for a factor (*van Exel & de Graaf, 2005*). Factors were then qualitatively interpreted based on the systematic and methodical approach to factor interpretation using the organization system described by *Watts & Stenner (2012)* (Chapter 7).

## RESULTS

All 11 participants completed the Q sort process and answered the open-ended interview questions following (completion rate 100%). Three study factors were identified following a by-person factor analysis of 11 Q sorts. Study factors were categorized as Factor A "Don't judge a book by its cover," Factor B "Gold standard or bust" and Factor C "The pragmatist(s)." Following varimax rotation, the three extracted factors explained 46.1% of the total variance in this study. All 11 Q sorts loaded significantly on one of the three factors and none of the Q sorts were confounded (meaning that none of the Q sorts loaded significantly on more than one factor). All three factors had eigenvalues greater than 1.00, however, only two of the factors had two or more significant Q sorts and can be considered exemplar individual factors. Table 3 shows the characteristics for the factors.

**Table 2  Q-set statements and factor array.**

| No. | Statements | Factors | | |
|---|---|---|---|---|
| | | A | B | C |
| 1 | The evidence from rapid reviews is good enough to inform low-risk, emergent policy or decision-making needs when the alternative is the use of no evidence | 1 | 1 | 0 |
| 2 | When time allows, a comprehensive systematic review of all available evidence should always be conducted | −1 | 2 | −3 |
| 3 | Deviating from accepted systematic review methods may introduce bias and impact the validity of the resulting rapid review, which may be an unacceptable risk for some for knowledge users | 0 | 3 | 1 |
| 4 | Further research comparing the methods and results of rapid reviews and systematic reviews is required before I decide how I feel about rapid reviews | −1 | 2 | 0 |
| 5 | Rapid reviews are too focused in scope and/or context to be generalizable to a variety of knowledge users | −2 | −2 | −1 |
| 6 | Rapid reviews mean different things to different people | 1 | 2 | 1 |
| 7 | Rapid reviews should only precede a more comprehensive and rigorous systematic review | −3 | −3 | −3 |
| 8 | The opportunity cost of a comprehensive SR or HTA is too high and it is more advantageous to conduct rapid reviews when timeliness is a factor | 0 | −2 | 1 |
| 9 | Rapid reviews do not replace SRs or HTAs | 0 | 2 | 0 |
| 10 | All evidence synthesis products, including rapid reviews, SRs, or HTAs, can be conducted very well or very poorly | 3 | 2 | 0 |
| 11 | Rapid reviews are comparable to SRs except they are done in a more timely fashion | −1 | 1 | 1 |
| 12 | Rapid reviews are 'quick and dirty' systematic reviews | −2 | 1 | −2 |
| 13 | Rapid reviews need to be tailored to the specific needs of the knowledge user | 0 | 2 | 3 |
| 14 | Rapid reviews meet the needs of knowledge users | 1 | 0 | 2 |
| 15 | There is a paucity of evidence on rapid reviews, so I cannot support or oppose their use in decision-making | −1 | 0 | −1 |
| 16 | There is so much overlap across the various evidence synthesis methods that I cannot generalize my opinion to favor one over the other without the context of the decision at hand | 0 | 0 | −1 |
| 17 | There is a risk involved in tailoring accepted SR methods to produce rapid reviews that we do not yet understand | −1 | 0 | 2 |
| 18 | Using rapid reviews to inform decisions is better than using no evidence at all | 2 | 0 | 0 |
| 19 | It is always appropriate to conduct a rapid review | −2 | −3 | −1 |
| 20 | Rapid reviews and all other evidence synthesis products hold the same value as long as they retain the core value of being transparent in conduct, include the highest quality evidence available and present results with a qualification on the strength of evidence | 2 | −1 | 0 |
| 21 | Appropriateness of a rapid review varies with the type of decision being made, and any financial, legal or other important contextual facets tied to the decision | 2 | 1 | 1 |
| 22 | My confidence in a rapid review is impacted by which methods are tailored to speed up the review process | 1 | 0 | 0 |
| 23 | My confidence in a rapid review is directly tied to results being presented and contextualized by the strength and applicability of the evidence | 0 | 0 | −1 |
| 24 | It is important to have minimum standards for the methodological conduct of rapid reviews | 0 | 1 | 2 |

| No. | Statements | Factors | | |
|---|---|---|---|---|
| | | A | B | C |
| 25 | It is important to have minimum standards for the reporting of rapid reviews (e.g., a PRISMA-RR) | 2 | 0 | 0 |
| 26 | Standardization of rapid review methods may conflict with the needs of knowledge users | 0 | −2 | 1 |
| 27 | The value of rapid reviews in the context of emergent decision-making needs outweighs the disadvantages or risk of bias and potentially 'imperfect' evidence | 1 | −1 | 2 |
| 28 | Knowledge users don't always need all of the evidence, they just need the best evidence to support their decision, and what is 'best evidence' is specific to the knowledge user | 1 | −2 | 3 |
| 29 | Knowledge users do not fully understand the implications of streamlining evidence synthesis methods to produce a more timely evidence product | 1 | 0 | 0 |
| 30 | Reporting of the results of rapid reviews must be tailored to the knowledge user(s) who commissioned the review | 0 | 0 | 0 |
| 31 | Rapid reviews that omit an assessment of the quality of included studies are useless to knowledge users | 0 | −1 | −1 |
| 32 | Rapid reviews can be timely and valid, even when methodological concessions are made | 1 | 1 | 1 |
| 33 | Transparency of process is more important than the actual methods used to produce rapid reviews, as transparency allows the end user to make their own assessment on validity and appropriateness | 0 | −2 | 2 |
| 34 | It is appropriate to endeavor to define a single, unique methodology for rapid reviews | −1 | −1 | −2 |
| 35 | Rapid reviews are not a unique methodology, they are simply a variation of a systematic review that can fall anywhere on the continuum of evidence synthesis methods | 1 | −1 | 0 |
| 36 | The results from a systematic review may not differ from those of a rapid review, but more research is needed to support this theory and quantify why results may be the same or different | 1 | 1 | 0 |
| 37 | I put more confidence in evidence produced in a systematic review than of a rapid review | −1 | 1 | −1 |
| 38 | The more time spent conducting the review of the evidence, the more valid the results of the review will be | −3 | −1 | −2 |
| 39 | Achieving a precise estimate of effect (from a SR) may not inform the decision-at-hand any better than a general estimate of effect (produced by a rapid review) | 2 | −1 | 1 |
| 40 | Rapid reviews should only be conducted when the alternate option is the use of no evidence to inform a decision | −2 | 1 | −1 |
| 41 | A well-conducted rapid review may produce better evidence than a poorly conducted systematic review | 3 | −1 | 2 |
| 42 | Any review of evidence that takes longer than three months to produce is not a rapid review | −1 | −2 | −1 |
| 43 | Any review of evidence that takes longer than one month to produce is not a rapid review | −2 | −1 | −2 |
| 44 | A rapid review must be justified with a valid rationale for both speeding up the process and tailoring rigourous methods for evidence synthesis | −1 | 0 | 0 |
| 45 | A good quality review of evidence is determined by the methods used, not by the speed at which it is completed | 2 | 1 | 0 |

(Continued)

| No. | Statements | Factors | | |
|---|---|---|---|---|
| | | **A** | **B** | **C** |
| 46 | It is difficult to tell a rapid review from a systematic review unless very specific nomenclature is used in the title or description of methods | −1 | 0 | 1 |
| 47 | A rapid review cannot be a systematic review | −2 | 3 | −2 |
| 48 | 'Rapid review' is too broad a phrase—doing a review in a more timely way can only be relative to how long it takes the same team to produce a full systematic review | 0 | −1 | −1 |
| 49 | Producers are more concerned with the methodology and validity of rapid reviews than knowledge users | 0 | 0 | 1 |
| 50 | It is difficult to judge the validity of a rapid review as the reporting is often truncated and protocols are not published | 0 | 0 | −2 |

**Note:**
Variance = 2.08, Standard Deviation = 1.4.

**Table 3** Q Factor characteristics.

| Characteristics | Factors | | |
|---|---|---|---|
| | **A** | **B** | **C** |
| Number of defining variables (n) | 8 | 2 | 1 |
| Composite reliability score | 0.97 | 0.889 | 0.8 |
| Standard error of factor scores | 0.174 | 0.333 | 0.447 |

The composite reliability coefficients ($r_c$) indicated construct validity for each factor as all values acceded the acceptable threshold of 0.70 (*Hair et al., 1998*).

Each factor, or salient perspective on rapid reviews, emerged from the attitudes and beliefs of the participating producers and knowledge users. Factors were 'named' according to their defining characteristics and following a careful, comprehensive interpretation of the factor arrays, scores and rankings. Participant profile information and results from the open-ended questions were also considered during interpretation. A description of each factor is presented with summary details of the participants who loaded significantly on the factor. Rankings of relevant items are provided. For example (+3) indicates that a statement is ranked in the +3 position which represents agreement in the factor array Q sort.

## Factor A. "Don't judge a book by its cover"

Factor A explained 24.5% of the total study variance. Eight of 11 participants significantly loaded on this factor. The majority of responses (87.5%) were from evidence producers.

This group was characterized by their view that we need to look more in-depth at the value or quality of individual review as opposed to a global assessment based on the labels traditionally employed to distinguish between evidence synthesis products (i.e. systematic review, rapid review). They had strong agreement (+3) with two statements in particular: "All evidence synthesis products, including rapid reviews, (systematic reviews, or health technology assessments), can be conducted very well or very poorly" and "A well-conducted rapid review may produce better evidence than a poorly

conducted systematic review." They similarly disagreed (−2) with the statement that "a rapid review cannot be a systematic review." Quality and value were again referenced in Factor A (+2) through participant agreement with the statement "Rapid reviews and all other evidence synthesis products hold the same value as long as they retain the core value of being transparent in conduct, include the highest quality evidence available and present results with a qualification on the strength of evidence." Participants defined by this factor may prescribe to a manifesto that acknowledges 'the good, the bad and the ugly' in all types of evidence synthesis products.

The relationship between time and quality were also common themes in Factor A. Participants agreed that value and quality were not tied to the length of time taken to complete a review, no matter how long or short. They agreed (+2) with the statement "A good quality review of evidence is determined by the methods used, not by the speed at which it is completed" and disagreed (−3) with the statement "The more time spent conducting the review of the evidence, the more valid the results of the review will be."

This group agreed (+2) that "Using rapid reviews to inform decisions is better than using no evidence at all" but that minimum reporting standards are desirable (+2) (e.g. A PRISMA statement for rapid reviews). This is supported by the disagreement (−2) documented for "Reporting of the results of rapid reviews must be tailored to the knowledge user(s) who commissioned the review."

## Factor B. "Gold standard or bust"

Factor B explained 9.6% of the total study variance. Two participants significantly loaded on this factor, one producer and one knowledge user.

This group strongly believed in the gold standard systematic review to meet the needs of knowledge-users, and that use of rapid reviews should be the exception, and not the rule. They firmly hold the belief (+3) that "deviating from accepted systematic review methods may introduce bias and impact the validity of the resulting rapid review, which may be an unacceptable risk for some for knowledge users" and that "rapid reviews cannot be systematic reviews." They also strongly disagreed (−3) that it is always appropriate to conduct rapid reviews. They agree (+2) with conducting a comprehensive systematic review of all available evidence when time allows, and that rapid reviews do not replace systematic reviews or health technology assessments. They were also clear in their disagreement (−2) with the statement "The opportunity cost of a comprehensive systematic review or health technology assessment is too high and it is more advantageous to conduct rapid reviews when timeliness is a factor" and generally agreed (+1) that "Rapid reviews should only be conducted when the alternate option is the use of no evidence to inform a decision." Participants also endorsed the view (+1) that "Rapids reviews are 'quick and dirty' systematic reviews," which participants in Factors A and C both disagreed with (−2). This sentiment is repeated in their disagreement (−2) with the principle suggested by "A well-conducted rapid review may produce better evidence than a poorly conducted systematic review."

Factor B, more than other groups, asserted that additional research in the area of rapid reviews is warranted. They also disagreed with statements pertaining to standardization of

rapid review methods conflicting with the needs of knowledge users. They are neutral in their beliefs that rapid reviews meet the needs of decision-makers, and strongly (−2) disagree with the idea that "Knowledge users don't always need all of the evidence, they just need the best evidence to support their decision, and what is 'best evidence' is specific to the knowledge user." To this group, it appears that a systematic review should always be considered 'best.'

### Factor C. "The pragmatist"

Factor C explained 11.9% of the total study variance. One participant, an evidence producer, significantly loaded on this factor.

This factor was characterized by a focus on the pragmatic needs of the knowledge user, balanced with the value of tailored rapid reviews and the inherent risk of bias that may accompany their use in decision-making processes. In opposition to those in Factor B, the participant felt strongly (+3) that "Knowledge users don't always need all of the evidence, they just need the best evidence to support their decision, and what is 'best evidence' is specific to the knowledge user." The evidence producer also strongly agreed (+2) that rapid reviews meet the needs of knowledge users and must be tailored to the individual specific needs of those commissioning the review (+3). The single participant disagreed (−3) that "When time allows, a comprehensive systematic review of all available evidence should always be conducted."

The Factor C viewpoint is also pragmatic in that the participant seemed to accept that use of rapid review approach may bring with it some risk. There is an emphasis in this viewpoint (+2) that "The value of rapid reviews in the context of emergent decision-making needs outweighs the disadvantages or risk of bias and potentially 'imperfect' evidence" and this is balanced with the requirement (+2) that transparency of process is important as it allows the end user to make their own assessment on validity and appropriateness. This is supported by the belief that there should be minimum standards for the methodological conduct of rapid reviews but disagreement (−2) that "It is appropriate to endeavor to define a single, unique methodology for rapid reviews." The evidence producer also admits they believe it is difficult to tell a rapid review from a systematic review unless it is explicitly stated (+1). This may carry forward the idea that although this individual likes to have the option of a pragmatic approach to meet their needs, they also require as much information about the methods used so they know how much confidence they can place on the results.

There was some overlap with the perspectives defined by Factor A. The participant firmly disagreed (−2) that "A rapid review cannot be a systematic review" and that "Rapid reviews are quick and dirty systematic reviews."

### Consensus and disagreement statements

Participants equally agreed or disagreed on several statements that were not distinguishable across factors, referred to as 'consensus statements' (Table 4). They generally agreed with the statement "Rapid reviews mean different things to different people." There was broad disagreement with three statements: "It is appropriate to

**Table 4 Statements showing agreement and disagreement across factors.**

| No. | Statements | Factor score | | |
|---|---|---|---|---|
| | | A | B | C |
| **Disagreement across factors** | | | | |
| 4 | Further research comparing the methods and results of rapid reviews and systematic reviews is required before I decide how I feel about rapid reviews | −1 | 2 | 0 |
| 3 | Deviating from accepted systematic review methods may introduce bias and impact the validity of the resulting rapid review, which may be an unacceptable risk for some for knowledge users | 0 | 3 | 1 |
| 2 | When time allows, a comprehensive systematic review of all available evidence should always be conducted | −1 | 2 | −3 |
| 47 | A rapid review cannot be a systematic review | −2 | 3 | −2 |
| 28 | Knowledge users don't always need all of the evidence, they just need the best evidence to support their decision, and what is 'best evidence' is specific to the knowledge user | 1 | −2 | 3 |
| **Agreement across factors** | | | | |
| 6 | Rapid reviews mean different things to different people | 1 | 2 | 1 |
| 7 | Rapid reviews should only precede a more comprehensive and rigorous systematic review | −3 | −3 | −3 |
| 34 | It is appropriate to endeavor to define a single, unique methodology for rapid reviews | −1 | −1 | −1 |
| 42 | Any review of evidence that takes longer than three months to produce is not a rapid review | −1 | −2 | −1 |
| 43 | Any review of evidence that takes longer than one month to produce is not a rapid review | −2 | −1 | −2 |

endeavor to define a single, unique methodology for rapid reviews," "Any review of evidence that takes longer than three months to produce is not a rapid review." and "Any review of evidence that takes longer than one month to produce is not a rapid review." All participants strongly disagreed with the statement "Rapid reviews should only precede a more comprehensive and rigorous systematic review." There was no concordance across factors when disagreement was considered; however, Factors A and C tended to disagree with statements in a similar pattern with varying degrees of magnitude.

# DISCUSSION

This research explored the attitudes and perceptions towards rapid reviews in a group of evidence producers and knowledge users. Analysis of the Q sorts identified three salient viewpoints which represent the broad spectrum of health care decision-makers and those who synthesize evidence to inform them. Factor A cautions against using labels to judge the quality or value or rapid reviews (or any other evidence synthesis products) and asserts that these variables should be assessed on an individual basis before appropriateness and worth can be gauged. Those prescribing to Factor B firmly hold the concepts of rigour and consistency of process found in a comprehensive systematic review true, and maintains that rapid reviews should be the exception and not the rule. Factor C has a focus on the pragmatic needs of the end-user instead of process, and is content balancing any risk that may be introduced by tailoring of methods with the imperative need for timely review of evidence. Importantly, results show that there are

quite opposing viewpoints on rapid reviews, but that some commonality across these perspectives also exists. The three factors described here are not necessarily exhaustive of the attitudes and perceptions held about rapid reviews; however, the relatively clear viewpoints may be valuable on their own as they offer insight into how and why various stakeholders act as they do and what issues may need to be resolved to increase uptake of evidence from rapid reviews. To our knowledge, this is the first study to specifically address this facet of rapid reviews.

Results of this study indicate that significant gaps still exist in perceived knowledge about rapid reviews. Most participants (Factor A, predominantly evidence producers) felt uncomfortable using the broad study labels to place any assessment of value or quality on the products falling under the nomenclature umbrella of 'rapid review.' Additionally, there was broad consensus across all of the factors extracted that the term 'rapid review' means different things to different people. The notion that all rapid reviews are not created equal is not a novel finding, but rather a commonly asserted trait consistent with previous reports (*Ganann, Ciliska & Thomas, 2010*; *Harker & Kleijnen, 2012*; *Khangura et al., 2012*; *Hailey, 2009*; *Merlin, Tamblyn & Ellery, 2014*). Our results indicate that most participants, regardless of viewpoint are well-aware of the heterogeneous range of approaches used to conduct rapid reviews, and that there is no standard or accepted way to carry out an accelerated evidence synthesis. In truth, it is still unclear whether rapid reviews should aspire to any standard at all. Participants in this study, a small sample of mostly evidence producers, stipulated that it is inappropriate to consider a single methodology for rapid reviews, which indicates that future studies should focus on more acutely describing the range of approaches captured by the term "rapid review" with the goal of providing some clarity to end-users. This requirement for multiple approaches has been put in practice by health technology assessment agencies like the CADTH and Health Quality Ontario who have evolved their internal rapid response and review programs into multi-product offerings. Recent endeavors by *Moher (2015)* and *Hartling et al. (2015)* suggest between four and seven functional groupings of rapid review exist and this provides a good basis for further study of the different approaches going forward. It is still uncertain if the ability to form an explicit definition for rapid reviews in hindered because of the varied approaches that must be captured by it.

Research can "both communicate and miscommunicate" according to *Glasziou et al. (2014)*, and research that is not adequately reported is at risk of becoming what the authors refer to as 'research waste' of time and resources. Transparency is central to the creation and evaluation of high quality research evidence, and something that rapid reviews are not well known for (*Harker & Kleijnen, 2012*). Providing sufficient information to end-users on research process and results is arguably more important for rapid reviews given that they are designed specifically to inform policy and practice and are based upon deviations from some accepted evidence synthesis practices. Results from analysis of the viewpoints in Factor A shows that there is a desire by researchers to make their own judgements on the potential value and quality of rapid review products; However, *Ganann, Ciliska & Thomas (2010)*, *Harker & Kleijnen (2012)* and most recently *Hartling et al. (2015)* have pointed out that inconsistencies in reporting and conduct

make it difficult for knowledge users to apply these judgements. The hesitancy of participants in factors A and B to fully endorse rapid review methodology without caveat suggests that transparency is a pressing factor that must be addressed. This raises the question of whether extensions to current reporting guidelines (e.g., PRISMA, PRISMA-P) or conduct checklists (e.g., AMSTAR) are desired, or required to encourage higher quality reporting. The opinions expressed in this study serve to remind researchers, academic and non-profit organizations, HTA agencies, and editors of journals that we need to do better when it comes to transparent reporting of research process and methods (*Shamseer et al., 2012*).

Knowledge users are faced with a challenge when it comes to rapid reviews. They must decide whether it is acceptable to trade-off the timely receipt of evidence with the risk that comprehensiveness of the end product may be compromised (*Featherstone et al., 2015*). We found that some stakeholders are accepting of this perceived trade-off (Factors A and C) than others (Factor B) when timely evidence sunthesis is required. Yet, there is little empirical evidence quantifying the impact of the methodological shortcuts used to expedite the review process (*Ganann, Ciliska & Thomas, 2010*; *Hailey et al., 2000*; *Khangura et al., 2012*; *Polisena et al., 2015*; *Watt et al., 2008a*; *Watt et al., 2008b*). Following the completion of this study, *Pham et al. (2016)* reported three agri-food case studies investigating the impact of applying methodological shorts-cuts to expedite the systematic review process. The shortcuts resulted in study omissions, a reduction in the number of cases where where meta-analysis was possible and generally less-precise pooled effect estimates. Despite this, *Hartling et al. (2015)* and *Polisena et al. (2015)* both point out that there are many well-established rapid response programs internationally which underscores that the risk associated with short-cuts is acceptable for certain end-users in specific situations. There may be situations where rapid reviews are inappropriate, as *Coates (2015)* and *Polisena et al. (2015)* suggest, such as those where there may be legal implications or where evidence is required to feed into the development of clinical practice guidelines (*Coates, 2015*). More research is warranted to clarify how decision-makers weigh these risks, to gauge when the risk of erroneous decision-making is too high, and which situations in particular are inappropriate for rapid review.

Factors extracted also showed opposing opinions on whether a rapid review can, in fact, be a systematic review. While it is clear that rapid reviews aspire to a standard, it is unclear what that standard actually is. Methodologically, we know that it is sometimes feasible for systematic processes to be sped up if resources are added. *Moher (2015)* provides a typology of rapid reviews that includes a category for 'traditional' systematic reviews done quickly. Due to poor reporting, we are often unable to tell whether a product like this has been tailored, or whether additional project resources were added to meet an expedited timeline. Theoretically a split exists amongst the various stakeholders too. Some fundamentally believe that if a product is a rapid review, by very nature then, it cannot also be systematic—which is equivalent to saying a rapid systematic review is an oxymoron, even though this term is often used for accelerated syntheses (*Schünemann & Moja, 2015*). Based on the results of the study by *Yuan & Hunt (2009)*, comparisons of rapid reviews to 'full,' 'traditional' or 'gold standard' systematic reviews

are common, and provide a notional frame of reference on which we can judge rapid review conduct, reporting and outcomes. We currently have little empirical evidence differentiating rapid reviews from systematic reviews, and no prospective research to-date has quantified differences between the two review types (*Tricco, Antony & Straus, 2015*; *Watt et al., 2008b*). Simply put, at this time we do not know when a systematic review becomes 'unsystematic.' Further, we may be relating rapid reviews back to a frame of reference which itself is flawed. Systematic reviews, with the possible exception of those carried out by the Cochrane collaboration, have been noted to have extreme variations in conduct, quality and reporting. Future research projects should aim to better quantify potential differences amongst these review types and to determine if assessing the quality and conduct of rapid reviews against this benchmark is fair or appropriate.

Another unique view stemming from this study relates to concepts of quality and time. Participants in Factor A specifically agreed that quality of a rapid review is not inherently tied to the time taken to complete the work, in contrast to the view of those in Factor C who held the opposite. Evidence actually supports the opinion of those in Factor C. *Harker & Kleijnen (2012)*, *Hartling et al. (2015)* and *Kelly, Moher & Clifford (2016)* have all examined samples of rapid reviews in depth, evaluated their quality and balanced this measure against the times taken to produce the reviews. Results have consistently shown a relationship between time and quality of reporting or conduct. Additional research is needed to confirm these findings across the typologies of rapid review approaches proposed by both Harker and Moher to investigate if time variables may be confounded by the approach.

## Strengths and limitations

The strengths in this study lie inherently with the methodology. Q methodology was selected for use in this project over other methods (e.g., simple questionnaire, interviews) because it offers the means to study subjective topics in a more systematic and rigorous manner. No other method explored enabled statements or qualitative descriptions to be quantified statistically using validated research techniques. Q methodology also offers a cost-effective way to potentially solicit opinion from a geographically diverse pool of evidence producers and knowledge users within the confines of the research project timeline.

It is important to acknowledge the potential limitations associated with this study. Although a large sample size is not required for a Q methodological study, results are based on a small number of participants who were predominantly evidence producers in Canada. No benefit was derived from efforts aimed at improving response rates, including sampling with replacement, use of user-friendly software accessible online or by portable devices (e.g., phone or tablet), and weekly email reminders. Due to funding limitations, we were unable to continue sampling with replacement until the desired study sample size of 50 was reached. The completion rate for those who participated was 100% so we do not consider the time-to-completion of 45 min to be a limiting factor; however, participant fatigue cannot be ruled out. The authors are aware of at least one other formal study on rapid reviews being administered at the time of our study, and two

other informal email surveys on the topic were also circulated within two months of this study. There would have been overlap in the evidence producers and users contacted, and this may have contributed to our low response rate.

Effort was made during concourse development to collect representative viewpoints from both users and producers of rapid reviews. Although the proportion of statements collected from each group was not empirically measured, the views of evidence producers may have been more common in the concourse or Q set. The number of evidence producers and knowledge users who participated in this study was disproportionate. Only two knowledge users completed the study, and their views and opinions are only expressed in a limited capacity in the Factors extracted. For this reason, our results are more reflective of the evidence producers view and do not fully capture knowledge users' perspectives or needs.

The smaller than desired sample size meant that certain methodological concessions had to be made when interpreting factors. Ideally, factors are defined by more than one Q sort, which we did not achieve for Factor C. While the same result may have been achieved with a larger sample size we cannot verify this claim with our study population. Although not intentional by design, this study population allows for some insight into the thought processes of a small sample of predominantly Canadian evidence producers. It is useful to keep in mind that evidence producers who participated in this study were not geographically diverse, and provided rankings based on their viewpoints tied to their own evidence synthesis products. Findings were interpreted by the study authors who, while they have much experience with knowledge users and the requirements of the healthcare decision-making processes, generally may identify more as evidence producers which could have unintentionally influenced the way factors were interpreted or results were presented.

Although we have identified a series of insightful viewpoints on rapid reviews, the range of viewpoints is not globally reflective of views of the wider population of evidence producers and stakeholders. Q methodology does not endeavour to make a claim of universal applicability or to represent the views of a larger sample (*Cross, 2005*; *Brown, 1993*). Q Methodology is also not intended to be a test of difference, and accordingly, results for evidence producers and knowledge users could not be compared and contrasted (*Hartling et al., 2015*). While we cannot exclude the influence of this dynamic on our results, historically, key memberships are not usually a defining influence on the generation of factors. In order to truly identify the views of researchers and decision-makers independently, two separate studies would need to be carried out using identical Q sets and procedures. Additionally, the comments received post-Q-sort more or less represent a unique, complementary qualitative study. Outside of interpreting the factors, they offer a depth of knowledge on this topic that deserves further exploration.

## CONCLUSIONS

This study has shown that there are distinct subsets of evidence producers and users who value and appreciate rapid reviews. At the same time, there are cautious segments of these populations who acknowledge the place of rapid reviews in evidence-informed decision-making under certain and exceptional circumstances. Much of the discourse in

this study revolved around central concepts of time and quality. While there is a growing body of evidence showing review quality decreases with abbreviated timeframes, there are still key stakeholders who believe that high-quality evidence can be synthesized in a timely manner. Empirical evaluation of the methodological implications of applying a rapid review approach are currently restricted to small case studies; therefore, a more fulsome study of these issues is necessary to explore whether there is evidence to support the particular views and opinions expressed in this Q methodology study. Research may be required to better define our gold standard of reference for rapid reviews before some of the uncertainties raised in dialogue by evidence producers and knowledge users can be resolved. Further study of evidence producer and user attitudes and opinions should be explored to evaluate whether the discourse changes as progress continues to exemplify methods and practice.

### Funding

Shannon E. Kelly received student financial support from the Canadian Agency for Drugs and Technologies in Health to sponsor the initiation of this study. David Moher and Tammy J. Clifford received no funding for this work. The funders had no role in study design, data collection and analysis, decision to publish, or preparation of the manuscript.

### Grant Disclosures

The following grant information was disclosed by the authors:
Canadian Agency for Drugs and Technologies in Health.

### Competing Interests

Tammy J. Clifford is an employee of the Canadian Agency for Drugs and Technologies in Health (CADTH).

Shannon E. Kelly and David Moher declare that they have no competing interests.

### Author Contributions

- Shannon E. Kelly conceived and designed the experiments, performed the experiments, analyzed the data, wrote the paper, prepared figures and/or tables, reviewed drafts of the paper, interpreted the analysis.
- David Moher conceived and designed the experiments, reviewed drafts of the paper, interpreted the analysis.
- Tammy J. Clifford conceived and designed the experiments, reviewed drafts of the paper, interpreted the analysis.

### Human Ethics

The following information was supplied relating to ethical approvals (i.e., approving body and any reference numbers):

This project was granted ethics approval (by letter) by the Ottawa Hospital Research Ethics Board May 25, 2012 under protocol #20120143-01H through Dr. David Moher.

## Data Deposition

The raw data has been supplied as Supplemental Dataset Files.

## Supplemental Information

Supplemental information for this article can be found online at http://dx.doi.org/10.7717/peerj.2522#supplemental-information.

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
