# Peer review of "Expediting evidence synthesis for healthcare decision-making: exploring attitudes and perceptions towards rapid reviews using Q methodology"

_PeerJ, doi:10.7717/peerj.2522_

## Round 0.1 · original submission · Minor Revisions

Please ensure you address all of the reviewers' comments in the revised article, and include a point by point description of the changes that have been made.

Reviewer 1 ·

Basic reporting

This manuscript on attitudes and perceptions on rapid reviews is well presented and clearly written. Only minor problems were noted. The authors have drawn on a range of relevant literature, though there are other publications on rapid reviews that might have been considered. The figures are relevant to the method used by the authors and of acceptable standard.

Experimental design

This was a primary study using Q methodology to obtain opinions from persons with interests in rapid reviews. 72: “The main goal of this study is to gauge how producers and knowledge-users feel about rapid reviews, and explore the range of opinion on this emerging evidence synthesis approach.”
Q methodology is well described. Possibly the material in 78-116 could be placed in the Introduction. A detailed description of the process used is given and is of interest.
120f: A summary of the sources used in defining the Q set are presented; possibly producers’ views would have been more common in some of these.
131: Perhaps the reference citations should be placed earlier in this section.
146: knowledge users
158 .. were industry representatives
Why were industry representatives excluded from the study? They might well have provided additional insights.
185 – 188. The issue of a small sample size is addressed here. However, there seems a possible difficulty when subsets are considered (see below).
210: (Figure 2)
214: What were the open ended questions, and how were the responses used?
217: Did the average completion time of 45 minutes cause any participant fatigue?
240-242: Factors were qualitatively interpreted by a single person. Was that interpretation checked?

Validity of the findings

266-267: Not clear to me if the proportions of gender are of great significance to the study findings, but there is a difficulty here in that 4 of 8 participants were female but there were only 3 male participants in the study (178, Table 1). Two other female participants are mentioned for Factor B (294) and one for Factor C (321). ……..
303 - 304: How were health technology assessments defined for the purposes of the study?
355: Is there a summary comment on the disagreements that could be added?
375: Rapid reviews are already widely used, is there a particular need to increase their uptake?
388: ..accurately??
404: “….may be based upon deviations from some accepted evidence synthesis practices.” would be more appropriate.
413 – 415: The opinions point in that direction but were more narrowly focused than the views in reference 31.
417-420: Reference 24 does not seem to include that message.
421- 422: I found this hard to follow. Is the sense that the effect of using a rapid review on the findings and their subsequent influence on decision makers is not known? Various examples of the outcomes of rapid reviews are included in the cited references.
430: …to Brassey
437 the most
430-442 Some interesting views from the cited presentation but unclear that all of these closely apply to the study.
483ff: Attention is drawn, appropriately, to the limitation of the small sample size. More emphasis might be given to disadvantage of having only two knowledge users.
493-494: That seems very optimistic…..
507ff: Some of the Conclusions section does not relate directly to the study findings.
518-519: The point on the need for improved reporting transparency and quality is well made, as earlier in the paper.
519-522: These views of ‘some participants’ do not seem to have been included earlier in the paper.

Additional comments

This is an interesting study, applying a fresh approach to obtaining opinions of different sectors on the place of rapid reviews.
The findings and opinions seem generally consistent with those included in several of the references, though that point is not made in the discussion.
The authors refer to rapid reviews as an “emerging evidence synthesis approach” (72) but such reviews have been widely used for many years, at least since the early 1990s, in Canada and other countries.
In general, the views and needs of the producers seem clearer than those of the users. I have the impression that the study as a whole, and the discussion details, do not fully capture knowledge users’ perspectives and needs.

·

Basic reporting

This article is interesting reading, and a good introduction to Q methodology to an audience that, I assume, are fairly new to it.

Experimental design

This article is interesting reading, and a good introduction to Q methodology to an audience that, I assume, are fairly new to it.

The results and conclusions are based on ranking conducted by 11 persons of which only two identify as evidence users. It should be made more clear in the results and discussion of results that these are mainly viewpoints and rankings by (a few) evidence producers of their own products! Sentences starting at line 378 and 387 could do with a reminder that these are views of evidence producers.

An explanation and discussion for why the total number of participants was 11 when the aim was 50, is warranted.

Validity of the findings

When producers of a pill or other product for sale present what they think of their product or rank information about it, there is often talk about Conflict of interests and potential Bias. In this case, eight of 11 participants, and probably all three of the authors are evidence producers – I think this article would improve if this issue of potential conflict of interest was raised and discussed.

On the same note, I think that statements about views and rankings from the evidence users perspective should be moderated to include some more uncertainty, there was only two – and they fell into different categories.


Line 481 “from a geographically diverse pool of evidence producers and knowledge users…” should be changed. Maybe it was geographically diverse for Canada – then say so. It cannot have been diverse for knowledge users with only two represented!

When reporting Factor C, would it be more precise to say she or one evidence producer rather than they? Participant rather than participants.

Line 453, “Based on the results of this study,….” Could be changed to Based on the results of the study by Yuan et al ,,…..

---

## Round 0.2 · accepted · Accept

All reviewers comments have been adaequately addressed.